# Assessment of the Potential of European Union Member States to Achieve Climate Neutrality

**Anna Bluszcz [1,\*], Anna Manowska [2] and Nur Suhaili Mansor [3]**

1   Department of Safety Engineering, Silesian University of Technology, 44-100 Gliwice, Poland
2   Department of Automatics and Industrial Informatics, Silesian University of Technology,
    44-100 Gliwice, Poland; anna.manowska@polsl.pl
3   Institute for Advanced and Smart Digital Opportunities, School of Computing, University Utara Malaysia,
    Sintok 06010, Kedah, Malaysia; nursuhaili@uum.edu.my
\*   Correspondence: anna.bluszcz@polsl.pl

**Abstract:** Climate neutrality is the main environmental goal set for the European Union Member States until 2050. EU economies can achieve this ambitious climate goal by reducing the emission intensity of economies, which has been achieved for many years by reducing pollution emitted by industry. The aim of the study is focused primarily on demonstrating the degree of relationship between the variables describing economic growth, GDP, and the level of $CO_2$ emissions. In the first stage of the research, the potential of countries to achieve climate neutrality was assessed, which was achieved by estimating the correlation between GDP indices in relation to 2013 and the level of $CO_2$ emissions. Research has shown that despite the countries' differences in the structure of their energy balances, they can achieve independence of economic growth from the emission level of their economies. The research also concerns Poland's special situation compared to other European Union countries according to energy balance based on coal. A model based on differential equations was used to simulate the impact of GDP, energy intensity, and the share of biofuels on temperature and $CO_2$ concentration until 2030, using data for Poland as an example. The aim of this analysis is to answer the question of whether the energy transformation in the country will achieve the assumed emission reduction goals by 2030.

**Keywords:** decarbonization; energy balance; climate neutrality; decoupling; $CO_2$; GDP

## 1. Introduction—Background of Climate Protection in the EU

Global problems related to climate change and the ongoing degradation of the natural environment related to the exploitation of mineral resources and the increasing consumption of goods and services constitute threats to maintaining the well-being of future generations. European Union countries are actively planning economic transformations to counteract the negative consequences of human activity. The implemented European Green Deal aims to transform the EU into a modern, resource-efficient, and competitive economy [1]. The sustainable development goals shaping the European Union's development policy are characterized in the Agenda 2030 adopted by all United Nations Member States in 2015 to ensure the well-being of citizens and care for the natural environment [2]. One of the most important factors influencing the globally observed climate change is greenhouse gas emissions, which in the European Union originate at 77.1% from the electricity production process in 2019 [3].

The EU has committed to a 2030 target to reduce its greenhouse gas emissions by at least 55% by the year 2030 compared to 1990 levels. This target was officially adopted as part of the "Fit for 55" package in July 2021. Key elements and goals of the "Fit for 55" package include elements such as:

- Greenhouse Gas Emission Reduction: The central objective is to cut greenhouse gas emissions by 55% by 2030 compared to 1990 levels. This involves various sectors, including energy, transport, industry, and buildings;
- Renewable Energy: The package aims to increase the share of renewable energy in the EU's energy mix, with a target of 40% of energy coming from renewable sources [4] by 2030;
- Energy Efficiency: There is a focus on improving energy efficiency across various sectors. The goal is to increase energy efficiency by 32.5%, contributing to overall emissions reduction;
- Carbon Market Expansion: The package proposes expanding the EU Emissions Trading System (EU ETS) to include new sectors and set more ambitious emission reduction targets;
- Transport Sector Transformation [5]: Measures are introduced to accelerate the shift to sustainable and low-emission transportation, including promoting electric vehicles and sustainable aviation fuels;
- Circular Economy [6] and Sustainable Agriculture: The package includes initiatives to promote a circular economy, reduce waste, and encourage sustainable agricultural practices;
- Social Aspects and Just Transition [7–9]: Recognising the social impacts of these changes, the package includes measures to ensure a fair and transition for workers and regions most affected by the transition to a greener economy.

One of the main goals of the "Fit for 55" package is the reduction of greenhouse gas emission levels, which is the main research area of the article. The main tool for monitoring emission levels and setting emission limits is the ETS System. The EU Emissions Trading System (ETS) aims to reduce carbon dioxide emissions in industry by obliging companies to have a permit for every Mg of $CO_2$ emitted. Companies must purchase permits through auctions. Additionally, the system offers certain incentives to stimulate innovation in the sector. The European Emissions Trading System is the world's first and largest market for carbon dioxide emission allowances. It regulates around 40% of the EU's total greenhouse gas emissions and covers around 10,000 power plants and production plants in the EU. To align the ETS with the emissions reduction targets of the European Green Deal, Parliament approved its update in April 2023. The reform includes reducing emissions in sectors covered by the ETS to 62% by 2030 (compared to 2005 levels). In 2023, the European Parliament approved the following provisions: Revision of the Emissions Trading Scheme (EU ETS) to cover polluting sectors such as buildings and road transport from 2027 (in ETS II) and maritime transport. The reforms are expected to phase out free aviation allowances by 2026 and will support the use of sustainable aviation fuels review of the Market Stability Reserve to address the structural imbalance between supply and demand for allowances in the EU ETS; implementation of the Carbon Leakage Instrument, which sets a greenhouse gas emission fee for imported goods to prevent relocation to countries with less ambitious climate targets; common an effort to reduce emissions between EU countries, increasing national emission reduction targets—in sectors not covered by the ETS, in particular transport, agriculture, construction and waste management—from 29% to 40% by 2030. Strengthening rules to increase carbon sequestration in land use and forestry sector (LULUCF) project to ensure that new passenger cars and commercial vehicles in the EU generate net zero emissions in 2035.

EU countries are making significant efforts to reduce greenhouse gas emissions, but the problem of related emissions remains with imports into the EU, which are constantly increasing, contributing to undermining the EU's internal efforts to reduce its global greenhouse gas footprint. It should be noted that net imports of goods and services into the EU currently represent 20% of the EU's domestic $CO_2$ emissions. Due to this fact, it is considered that there is an urgent need to better monitor them in order to identify possible measures to reduce global greenhouse gas emissions in the EU. The consequences of introducing emission limits in the European Union have a negative impact on the level of

competitiveness of domestic products, taking into account environmental costs. Based on data presented on the WITS World Integrated Trade Solution website, in the years 2000 to 2017, all European Union countries recorded an increase in imports of at least 100% to 250%. This fact means that the European Union imports goods produced in countries outside the zone of applicable greenhouse gas emission limits. This phenomenon is very negative as a consequence of the so-called carbon leakage phenomenon, which means the relocation of energy-intensive industries to places where emission limits do not apply. Importing goods from other regions of the world, including Asia, causes negative environmental consequences in areas where rigorous environmental protection standards are not applied. European Union countries have taken active steps to counteract this phenomenon. In order not to limit the competitiveness of companies covered by the EU Emissions Trading System (EU ETS), less restrictive rules for the allocation of free allowances apply to sectors exposed to the risk of carbon leakage. This is to reduce the risk of moving their production to countries where the industry has no restrictions on carbon dioxide emissions. In 2009, the EU Commission published the first list of sectors and subsectors exposed to the risk of flight, which included, among others, mining and enrichment of hard coal, mining of minerals for the chemical industry, and the production of fertilizers, copper and other non-ferrous metals, extraction of crude oil and natural gas, and many others [10]. The next approach is to implement changes as part of the "Fit for 55" package across all relevant policy instruments, which includes, in particular, the review of sectoral legislation in the fields of climate, energy, transport, and taxation. One of the elements of this package is the $CO_2$ emission border adjustment mechanism ("CBAM") announced in the European Green Deal. Carbon border adjustment mechanism (CBAM). The idea is to impose a carbon price on imported goods based on the emissions associated with their production. This mechanism aims to ensure that domestic industries remain competitive and do not face a disadvantage against foreign competitors who may not be subject to similar carbon pricing. In March 2021, the European Parliament adopted a resolution supporting the introduction of a WTO-compliant border price adjustment mechanism considering $CO_2$ emissions. As outlined in the European Green Deal, CBAM will ensure that the prices of imported products reflect the emissions associated with their production more closely. This measure is designed to be consistent with the rules of the World Trade Organization and the Union's other international obligations [11].

The EU Emissions Trading System (ETS) clearly affects the reduction of greenhouse gas emission levels, particularly in the energy, manufacturing, and aviation sectors. These revenues provide the opportunity to support climate and energy action while complementing both the Innovation and Modernization funds managed by the European Commission to support the energy and industrial transition. Additionally, in 2023, an agreement was reached on the creation of ETS2, which will cover fuel combustion in buildings, road transport, and other sectors.

Figure 1 presents the effects of the ETS on individual EU countries. The chart shows the sum of Member States' revenues from auctioning ETS emission allowances (in EUR a million) and the corresponding amounts spent on climate action and climate energy goals (in EUR a million) in the years 2013 to 2022. As can be seen, the countries with the highest level of revenues in the ETS system are the main energy producers in the EU, i.e., Germany, Poland, Italy, Spain, France, Greece, and The Netherlands. In the countries mentioned, the revenues obtained were spent on climate goals in the following shares: Germany 100%, Poland 50%, Italy > 50%, Spain 77%, France 91%, Greece 100%, and The Netherlands (reported amount 52%; investments included in other climate expenditure during over 100%). Among the member states, the lowest share of expenditure on climate goals in the revenues achieved was achieved by Slovakia (21%), Portugal (28%), and Romagna (35%). The remaining countries show levels higher than 50%, which should be assessed positively. Based on the presented data, we can see the direct impact of the ETS system on climate protection activities and changes in the energy systems of EU countries.

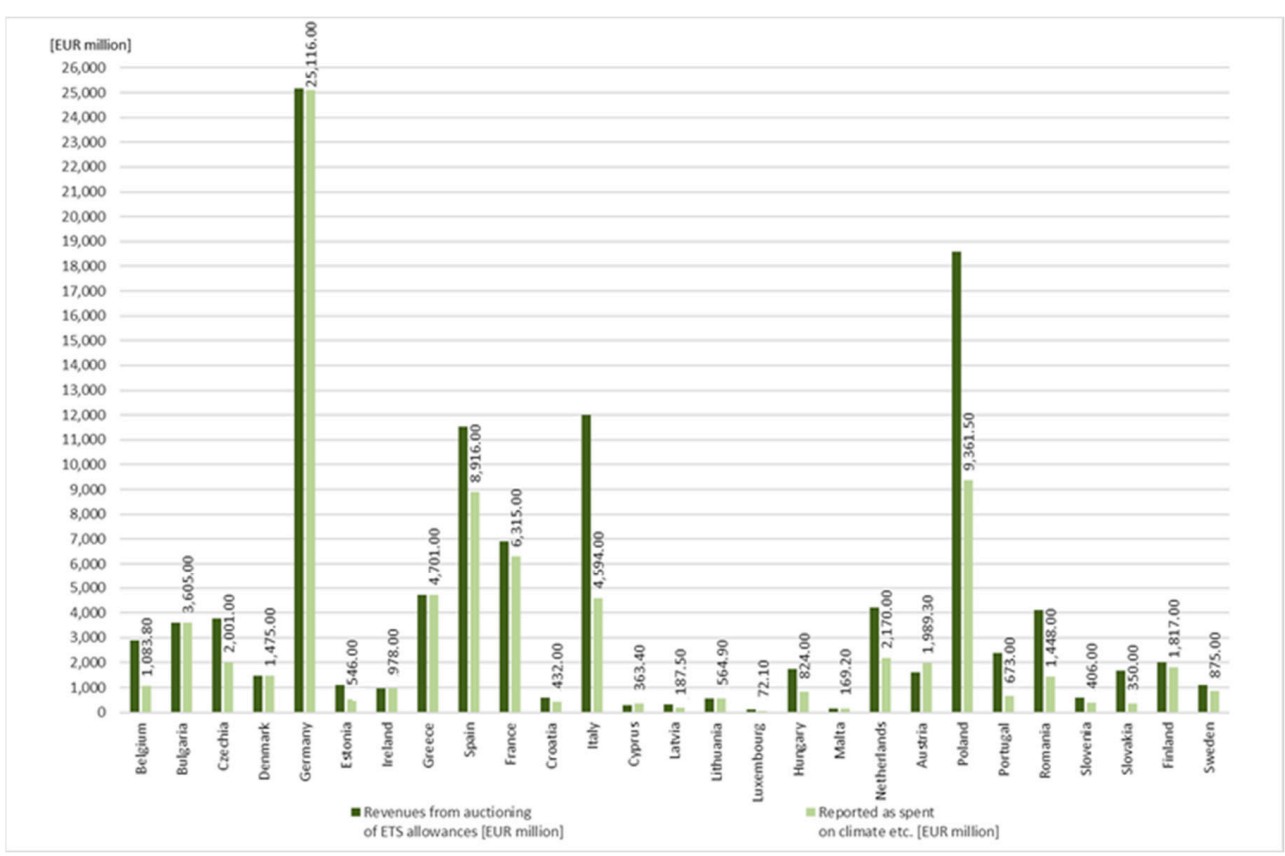

**Figure 1.** Revenues from ETS system, source: own study based on [12].

The long-term goal of the European Union 2050 target is to achieve climate neutrality by the year 2050. Climate neutrality, also known as net-zero emissions, means that any remaining greenhouse gas emissions are balanced by removing an equivalent amount of emissions from the atmosphere or offsetting through other means. Achieving climate neutrality implies a comprehensive transformation of various sectors [13], including energy, transportation, industry, and agriculture. It is important to note that these targets and commitments are part of the EU's broader climate and energy policy framework. The specific policies and measures to achieve these targets are outlined in various legislative proposals and initiatives, such as the "Fit for 55" package mentioned earlier. Additionally, countries within the EU may have individual targets and plans to contribute to the overall EU goals. The success of these targets depends on the effective implementation of policies, investments in sustainable technologies, and international cooperation to address global climate challenges.

Achieving climate goals is directly related to the transformation of energy markets in European Union countries, which is aimed primarily at increasing the share of renewable sources in energy balances, diversifying sources in order to reduce the degree of dependence on imported fuels, and reducing greenhouse gas emission levels through the implementation of low-emission solutions [14].

## 2. Study Method

The aim of the research is to assess the potential of member countries to achieve climate neutrality in the future in accordance with the adopted EU environmental goals by 2050. In order to carry out this study, analyses were carried out in two stages.

The first stage of the research concerned the analysis of economic growth levels in 2013–2022 and the emission levels of economies simultaneously while also determining the potential of economies to transform to achieve climate neutrality.

The validity of this research area is very significant because it allows examining the trend in meeting the requirements for reducing $CO_2$ emission levels while maintaining society's current standard of living. The assessment of the economies' potential was made on the basis of the Pearson correlation index between the GDP and $CO_2$ indices in subsequent years up to the base year of 2013. The positive correlation index indicates a linear relationship for the studied variables and is assessed as the least desirable phenomenon, as it means the dependence of economic growth on the emission level. We assess the negative level of dependence as a positive phenomenon in assessing the economy's potential to achieve climate neutrality. The research results clearly showed that the member countries are significantly diversified, but they are on the path of energy transformation and show a high potential to achieve energy neutrality by 2050.

The second part of the research in the article concerns Poland's special situation compared to other European Union countries. In this part of the research, a model based on differential equations was used to simulate the impact of GDP, energy intensity, and the share of biofuels on temperature and $CO_2$ concentration until 2030, using the example of data for Poland. The aim of this analysis is to answer the question of whether the energy transformation in the country will achieve the assumed emission reduction goals by 2030.

## 3. Results of Analysis

Social prosperity, which is associated with a constant increase in the standard of living, clearly affects the increase in greenhouse gas emissions, especially carbon dioxide ($CO_2$), produced as a result of the combustion of energy fuels [15]. The member states of the European Union differ significantly in terms of energy balances [16]. Aspects related to the energy mix are of key importance in the energy transformation process of member countries, and therefore, the criteria in striving to meet environmental goals should be adapted to the financial capabilities (economic development) of each country and the energy balance [17], the transformation of which is a process that can be implemented in the long -term period [18–28]. The dynamic economic growth observed on a global scale since the Industrial Revolution has had a negative effect in the form of an increase in greenhouse gas emissions. The direction enabling economic growth to be maintained while respecting the natural environment is to achieve the so-called "decoupling" of gross domestic product (GDP) from the level of greenhouse gas emissions. In relation to climate change, the environmental pressure to be decoupled from economic growth is greenhouse gas (GHG) emissions, particularly carbon dioxide ($CO_2$), the main gas responsible for the greenhouse effect. Decoupling is therefore defined in the context of climate change by the Intergovernmental Panel on Climate Change (IPCC) as economic growth that is no longer strongly associated with the consumption of fossil fuels, as these are the primary sources of $CO_2$ emissions [20].

This is the main research area of the article, in which the following hypothesis is put forward:

**Hypothesis 1 (H1):** *European Union countries have the potential to ensure sustainable development while maintaining the growth of society's well-being.*

In order to answer the research problem defined in this way, data were collected on the emission levels of carbon dioxide [Mg] and Gross domestic product at market price (Current prices, euro per capita) for all European Union countries. In the next step of the analysis, the actual GDP and $CO_2$ statistical data were transformed in subsequent years from 2014 to 2022 into an index in relation to the base year 2013. The results are presented in Table 1.

**Table 1.** Indexes of GDP and CO$_2$ and Pearson correlation between GDP and CO$_2$.

| Index of Gross Domestic Product at Market Prices to 2013 | | | | | | | | | | |
|---|---|---|---|---|---|---|---|---|---|---|
| | **2013** | **2014** | **2015** | **2016** | **2017** | **2018** | **2019** | **2020** | **2021** | **2022** |
| Belgium | 1.0000 | 1.0210 | 1.0497 | 1.0781 | 1.1110 | 1.1434 | 1.1832 | 1.1341 | 1.2451 | 1.3471 |
| Bulgaria | 1.0000 | 1.0294 | 1.1019 | 1.1813 | 1.2815 | 1.3817 | 1.5233 | 1.5337 | 1.7841 | 2.2919 |
| Czechia | 1.0000 | 0.9888 | 1.0600 | 1.1068 | 1.2083 | 1.3085 | 1.3942 | 1.3296 | 1.4680 | 1.7040 |
| Denmark | 1.0000 | 1.0215 | 1.0423 | 1.0720 | 1.1093 | 1.1319 | 1.1542 | 1.1586 | 1.2709 | 1.3980 |
| Germany | 1.0000 | 1.0370 | 1.0628 | 1.0921 | 1.1340 | 1.1644 | 1.1994 | 1.1741 | 1.2473 | 1.3270 |
| Estonia | 1.0000 | 1.0642 | 1.0971 | 1.1543 | 1.2654 | 1.3729 | 1.4735 | 1.4413 | 1.6362 | 1.8883 |
| Ireland | 1.0000 | 1.0825 | 1.4460 | 1.4635 | 1.6016 | 1.7359 | 1.8634 | 1.9415 | 2.2285 | 2.5506 |
| Greece | 1.0000 | 0.9915 | 0.9933 | 0.9866 | 1.0024 | 1.0195 | 1.0420 | 0.9397 | 1.0396 | 1.1913 |
| Spain | 1.0000 | 1.0146 | 1.0602 | 1.0949 | 1.1401 | 1.1757 | 1.2068 | 1.0785 | 1.1785 | 1.2907 |
| France | 1.0000 | 1.0106 | 1.0293 | 1.0421 | 1.0670 | 1.0923 | 1.1216 | 1.0623 | 1.1431 | 1.2017 |
| Croatia | 1.0000 | 0.9943 | 1.0296 | 1.0860 | 1.1585 | 1.2350 | 1.3095 | 1.1929 | 1.4107 | 1.6619 |
| Italy | 1.0000 | 1.0090 | 1.0277 | 1.0550 | 1.0823 | 1.1062 | 1.1249 | 1.0453 | 1.1526 | 1.2334 |
| Cyprus | 1.0000 | 0.9799 | 1.0115 | 1.0669 | 1.1290 | 1.1902 | 1.2556 | 1.1830 | 1.3230 | 1.4539 |
| Latvia | 1.0000 | 1.0468 | 1.0981 | 1.1440 | 1.2279 | 1.3366 | 1.4117 | 1.3993 | 1.5645 | 1.8207 |
| Lithuania | 1.0000 | 1.0532 | 1.0852 | 1.1443 | 1.2616 | 1.3713 | 1.4785 | 1.5055 | 1.6970 | 2.0084 |
| Luxembourg | 1.0000 | 1.0302 | 1.0521 | 1.0661 | 1.0792 | 1.0944 | 1.1144 | 1.1327 | 1.2511 | 1.3137 |
| Hungary | 1.0000 | 1.0426 | 1.1094 | 1.1471 | 1.2565 | 1.3475 | 1.4521 | 1.3688 | 1.5353 | 1.6883 |
| Malta | 1.0000 | 1.0788 | 1.2038 | 1.2402 | 1.3662 | 1.4413 | 1.5185 | 1.3877 | 1.5845 | 1.7630 |
| Netherlands | 1.0000 | 1.0132 | 1.0364 | 1.0583 | 1.0964 | 1.1430 | 1.1929 | 1.1621 | 1.2634 | 1.3779 |
| Austria | 1.0000 | 1.0204 | 1.0440 | 1.0709 | 1.0992 | 1.1408 | 1.1709 | 1.1180 | 1.1848 | 1.2929 |
| Poland | 1.0000 | 1.0466 | 1.1080 | 1.0951 | 1.2012 | 1.2874 | 1.3746 | 1.3598 | 1.4965 | 1.7146 |
| Portugal | 1.0000 | 1.0209 | 1.0644 | 1.1080 | 1.1669 | 1.2239 | 1.2785 | 1.1945 | 1.2877 | 1.4436 |
| Romania | 1.0000 | 1.0573 | 1.1315 | 1.1888 | 1.3301 | 1.4797 | 1.6168 | 1.5986 | 1.7664 | 2.0993 |
| Slovenia | 1.0000 | 1.0311 | 1.0638 | 1.1068 | 1.1763 | 1.2508 | 1.3141 | 1.2638 | 1.4011 | 1.5277 |
| Slovakia | 1.0000 | 1.0240 | 1.0741 | 1.0872 | 1.1315 | 1.1991 | 1.2587 | 1.2435 | 1.3394 | 1.4520 |
| Finland | 1.0000 | 1.0083 | 1.0266 | 1.0535 | 1.0934 | 1.1264 | 1.1562 | 1.1456 | 1.2039 | 1.2856 |
| Sweden | 1.0000 | 0.9835 | 1.0100 | 1.0211 | 1.0372 | 1.0052 | 1.0080 | 1.0087 | 1.1280 | 1.1667 |

| Index of Air pollutants and greenhouse gases; Carbon dioxide [Mg] to 2013 | | | | | | | | | | | Correlation Pearson index GDP and CO$_2$ |
|---|---|---|---|---|---|---|---|---|---|---|---|
| | **2013** | **2014** | **2015** | **2016** | **2017** | **2018** | **2019** | **2020** | **2021** | **2022** | |
| Belgium | 1.0000 | 0.9675 | 1.0052 | 1.0006 | 1.0005 | 1.0255 | 1.0342 | 0.9461 | 0.9561 | 0.9115 | −0.5189 |
| Bulgaria | 1.0000 | 1.0662 | 1.1303 | 1.0678 | 1.1198 | 1.0199 | 0.9888 | 0.8445 | 0.9840 | 1.0532 | −0.2610 |
| Czechia | 1.0000 | 0.9907 | 0.9780 | 1.0030 | 0.9853 | 1.0013 | 0.9430 | 0.8347 | 0.8933 | 0.8959 | −0.6751 |
| Denmark | 1.0000 | 0.9166 | 0.9202 | 0.9604 | 0.9636 | 0.9993 | 0.9553 | 0.8176 | 0.8947 | 0.8481 | −0.5798 |
| Germany | 1.0000 | 0.9425 | 0.9545 | 0.9487 | 0.9340 | 0.9061 | 0.8200 | 0.7202 | 0.7813 | 0.7719 | −0.8307 |
| Estonia | 1.0000 | 0.9542 | 0.7938 | 0.8481 | 0.9136 | 0.8550 | 0.5683 | 0.4042 | 0.4649 | 0.5538 | −0.7855 |
| Ireland | 1.0000 | 1.0183 | 1.0719 | 1.1914 | 1.2483 | 1.2691 | 1.2383 | 0.8606 | 0.9276 | 1.1421 | 0.0106 |
| Greece | 1.0000 | 0.9644 | 0.9169 | 0.8922 | 0.9608 | 0.9566 | 0.8858 | 0.7462 | 0.7662 | 0.7496 | −0.3706 |
| Spain | 1.0000 | 1.0146 | 1.0907 | 1.0312 | 1.1056 | 1.0705 | 0.9886 | 0.8208 | 0.8798 | 0.9284 | −0.1873 |
| France | 1.0000 | 0.9205 | 0.9237 | 0.9230 | 0.9587 | 0.9335 | 0.9134 | 0.8020 | 0.8681 | 0.8540 | −0.5290 |
| Croatia | 1.0000 | 0.9527 | 0.9423 | 0.9468 | 0.9920 | 0.9226 | 0.9358 | 0.8817 | 0.8979 | 0.8433 | −0.7899 |
| Italy | 1.0000 | 0.9388 | 0.9601 | 0.9505 | 0.9546 | 0.9350 | 0.9121 | 0.8005 | 0.8919 | 0.9035 | −0.3160 |
| Cyprus | 1.0000 | 1.0670 | 1.0291 | 1.0978 | 1.1440 | 1.1510 | 1.1096 | 1.0664 | 1.0646 | 1.0679 | 0.2695 |
| Latvia | 1.0000 | 0.9765 | 1.0187 | 0.9963 | 1.0124 | 1.1021 | 1.0912 | 0.8801 | 0.9246 | 0.8666 | −0.4678 |
| Lithuania | 1.0000 | 1.0599 | 1.1124 | 1.0836 | 1.1420 | 1.2431 | 1.2754 | 1.4236 | 1.2851 | 1.1960 | 0.6481 |
| Luxembourg | 1.0000 | 1.0332 | 1.1627 | 1.1386 | 1.1351 | 1.1563 | 1.2521 | 1.1157 | 1.1078 | 1.0610 | 0.0422 |
| Hungary | 1.0000 | 1.0124 | 1.0666 | 1.0562 | 1.1130 | 1.1237 | 1.0848 | 1.0002 | 1.0168 | 0.9860 | −0.0961 |
| Malta | 1.0000 | 1.0002 | 0.6936 | 0.5566 | 0.6271 | 0.6465 | 0.6929 | 0.6002 | 0.5894 | 0.6299 | −0.6921 |
| Netherlands | 1.0000 | 1.0006 | 1.0365 | 1.0319 | 1.0148 | 0.9902 | 0.9697 | 0.8475 | 0.8544 | 0.8086 | −0.8519 |
| Austria | 1.0000 | 0.9483 | 0.9712 | 0.9556 | 1.0178 | 0.9657 | 1.0365 | 0.9102 | 0.9984 | 0.9342 | −0.0818 |
| Poland | 1.0000 | 0.9736 | 0.9849 | 1.0259 | 1.0788 | 1.0750 | 1.0284 | 0.9703 | 1.0850 | 1.0350 | 0.4019 |
| Portugal | 1.0000 | 0.9968 | 1.0984 | 1.0477 | 1.1627 | 1.0821 | 0.9911 | 0.8228 | 0.8012 | 0.8340 | −0.5281 |
| Romania | 1.0000 | 0.9963 | 0.9784 | 0.9542 | 0.9858 | 0.9845 | 0.9315 | 0.8760 | 0.8995 | 0.8193 | −0.8952 |
| Slovenia | 1.0000 | 0.8778 | 0.8933 | 0.9355 | 0.9539 | 0.9787 | 0.9648 | 0.9199 | 0.9122 | 0.8571 | −0.3245 |
| Slovakia | 1.0000 | 0.9526 | 0.9708 | 0.9799 | 1.0076 | 1.0118 | 0.9245 | 0.8355 | 0.9441 | 0.8340 | −0.6808 |
| Finland | 1.0000 | 0.9134 | 0.8471 | 0.9144 | 0.8592 | 0.8871 | 0.8314 | 0.6922 | 0.6994 | 0.6845 | −0.8469 |
| Sweden | 1.0000 | 0.9643 | 0.9760 | 1.0060 | 0.9702 | 0.9666 | 0.9410 | 0.8118 | 0.8699 | 0.8487 | −0.5718 |

[own elaboration].

Based on data tab 1, it is possible to demonstrate whether the phenomenon of decoupling occurred in the analyzed period for the member countries. In order to obtain greater transparency and readability of the data regarding the decoupling phenomenon, the data from Table 1 is presented in Figures 2–6. Due to the fact that the highest shares of emissions come from the electricity production process, the decoupling analysis was prepared based on the procedure of classifying countries in terms of their similarity in energy balances.

Based on previous research in the work by Bluszcz and Manowska [21], member countries were classified according to criteria closely related to the countries' energy balances, energy consumption, and energy efficiency. Nine classification criteria were selected and used in the grouping, which included:

-   Consumption of electric Energy generated from renewables per capita (TWH/person);
-   Hard coal consumption (million tons/person),
-   Greenhouse gas emissions per capita;
-   Available for final consumption: Gigawatt-hour per person;
-   Final energy consumption of thousand tons of oil equivalent (TOE) per person;
-   Petroleum available for final consumption (Gigawatt-hour);
-   Natural gas (Terajoule gross calorific value—GCV) per person;
-   Energy intensity of GDP (Kilograms of oil equivalent (KGOE) per thousand euro;
-   Import dependency%.

On the basis of these variables, five clusters of countries similar in terms of energy balances were identified, which were used in this analysis to identify the phenomenon of decoupling in the studied clusters.

Cluster 1: Sweden, Finland, and Luxembourg.
Cluster 2: Romania, Latvia, Slovenia, and France.
Cluster 3: Italy, Portugal, Spain, Hungary, Croatia, Lithuania, and Greece.
Cluster 4: Estonia, Bulgaria, Poland, Slovakia, and Czechia.
Cluster 5: Austria, Denmark, Ireland, Germany, The Netherlands, and Belgium.

Multidimensional analysis is a very useful tool for organizing and grouping objects in clusters, which are described by multiple criteria expressed using different units of measurement. The results of these studies constitute the basis for further comparative analyses carried out in groups of countries similar in terms of selected energy criteria.

In the first phase of the analysis, data on GDP and $CO_2$ level indices were presented for the years 2013–2022. The data were presented according to the selected five clusters of energy-similar countries in Figures 2–6.

In Figures 2–6, the $Y$ axis is the index of the GDP and $CO_2$ values of each subsequent year to the value in 2013, i.e., GDP2014/GDP2013 etc., i.e., $CO_2$ 2014/$CO_2$ 2013, etc.

I GDP = GDP i/GDP x
I GDP = Index of GDP
i = years from 2013 to 2022
x = year 2013

In Figures 1–5, the $Y$ axis is the index of the $CO_2$ value of each subsequent year to the value in 2013, i.e., $CO_2$ 2014/$CO_2$ 2013, etc.

I $CO_2$= $CO_2$ i/$CO_2$ x
I GDP = Index of $CO_2$
i = years from 2013 to 2022
x = year 2013

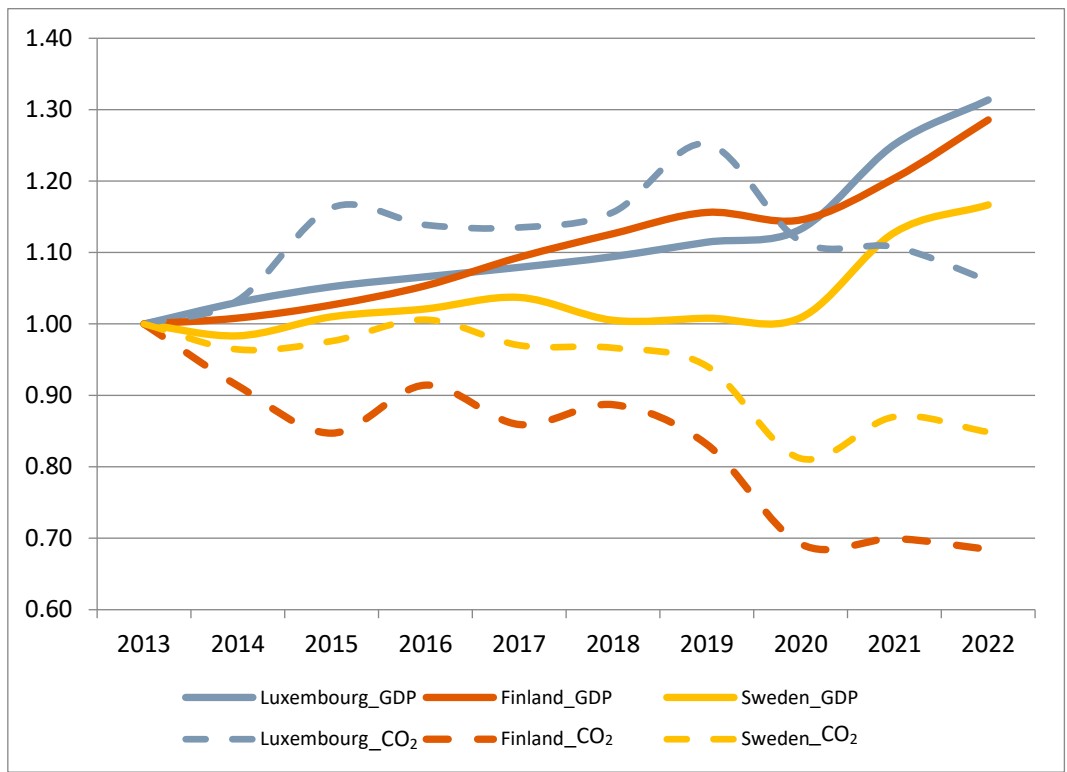

**Figure 2.** Analysis of changes in the level of GDP and $CO_2$ indices in the years 2013–2022 up to the base year 2013—cluster 1, source: own study.

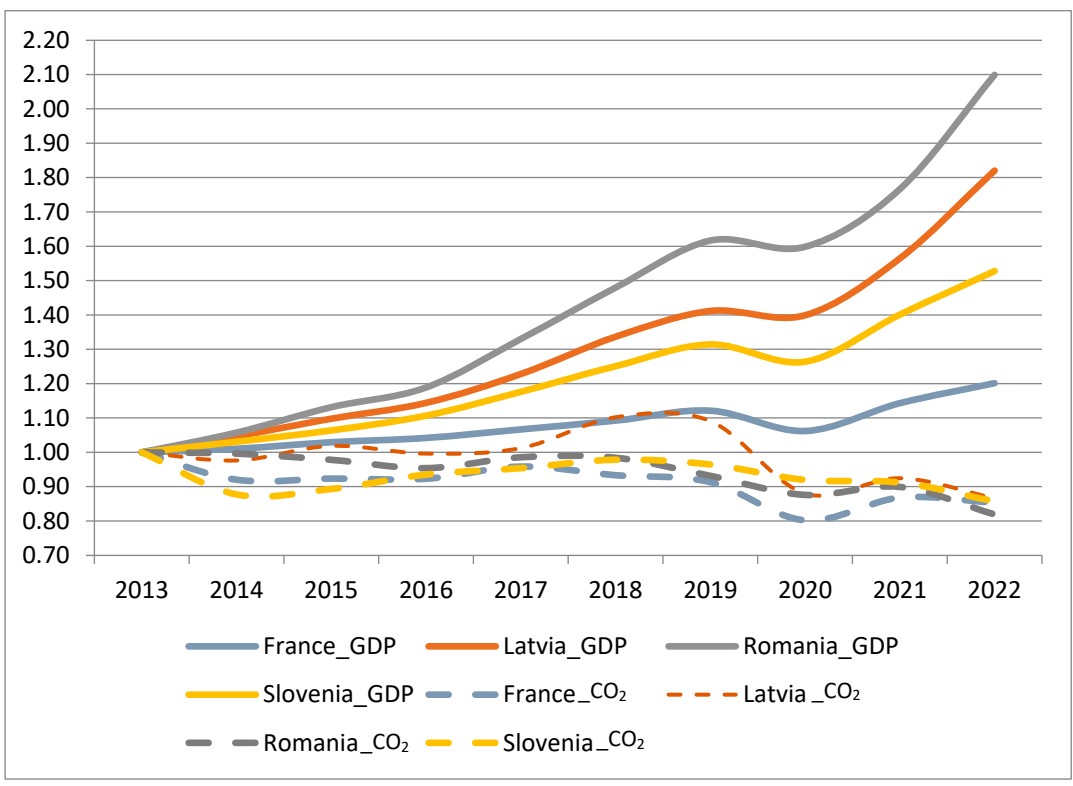

**Figure 3.** Analysis of changes in the level of GDP and $CO_2$ indices in the years 2013–2022 up to the base year 2013—cluster 2, source: own study.

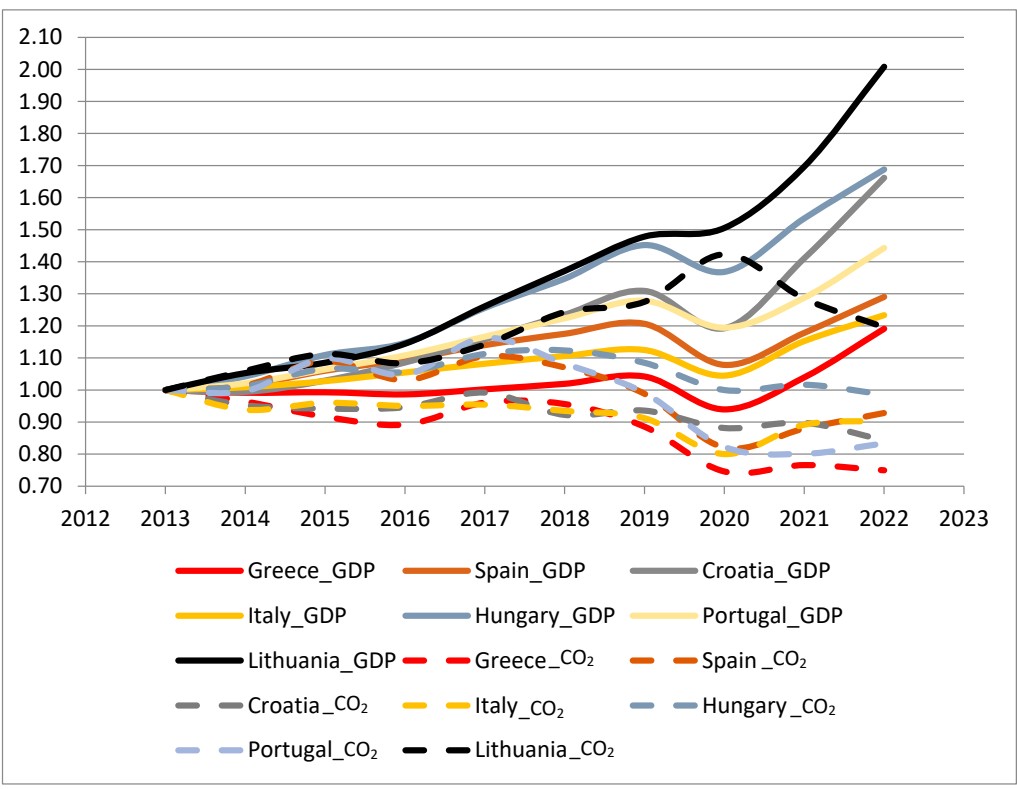

**Figure 4.** Analysis of changes in the level of GDP and CO$_2$ indices in the years 2013–2022 up to the base year 2013—cluster 3, source: own study.

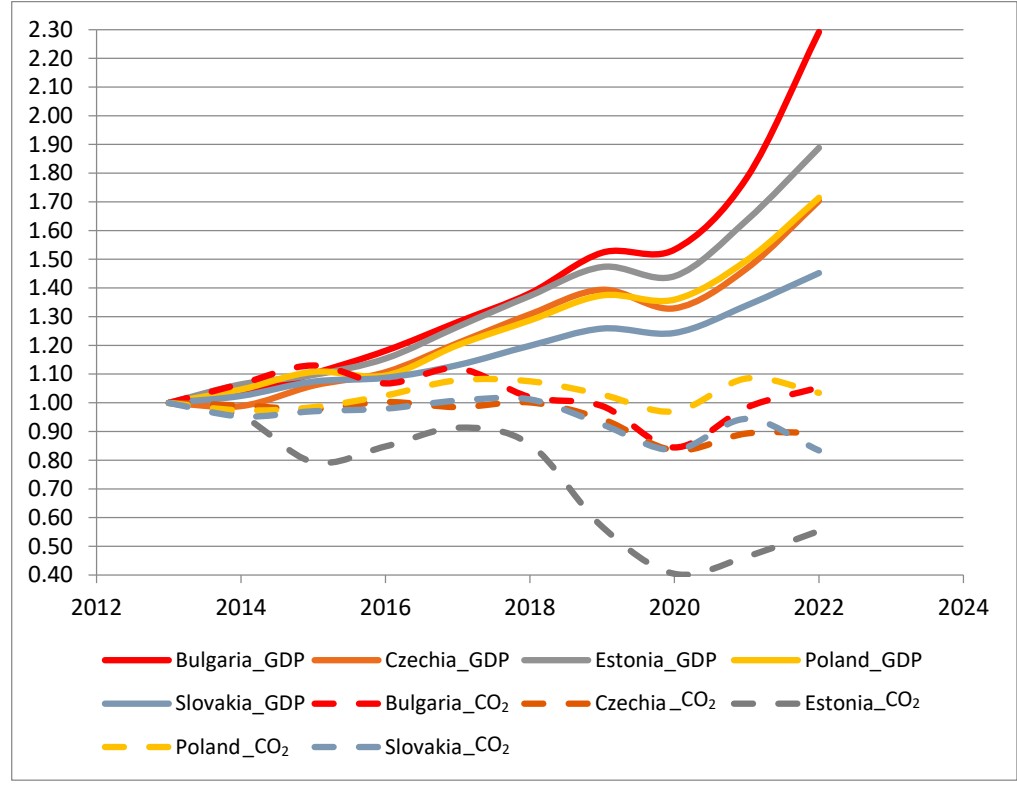

**Figure 5.** Analysis of changes in the level of GDP and CO$_2$ indices in the years 2013–2022 up to the base year 2013—cluster 4, source: own study.

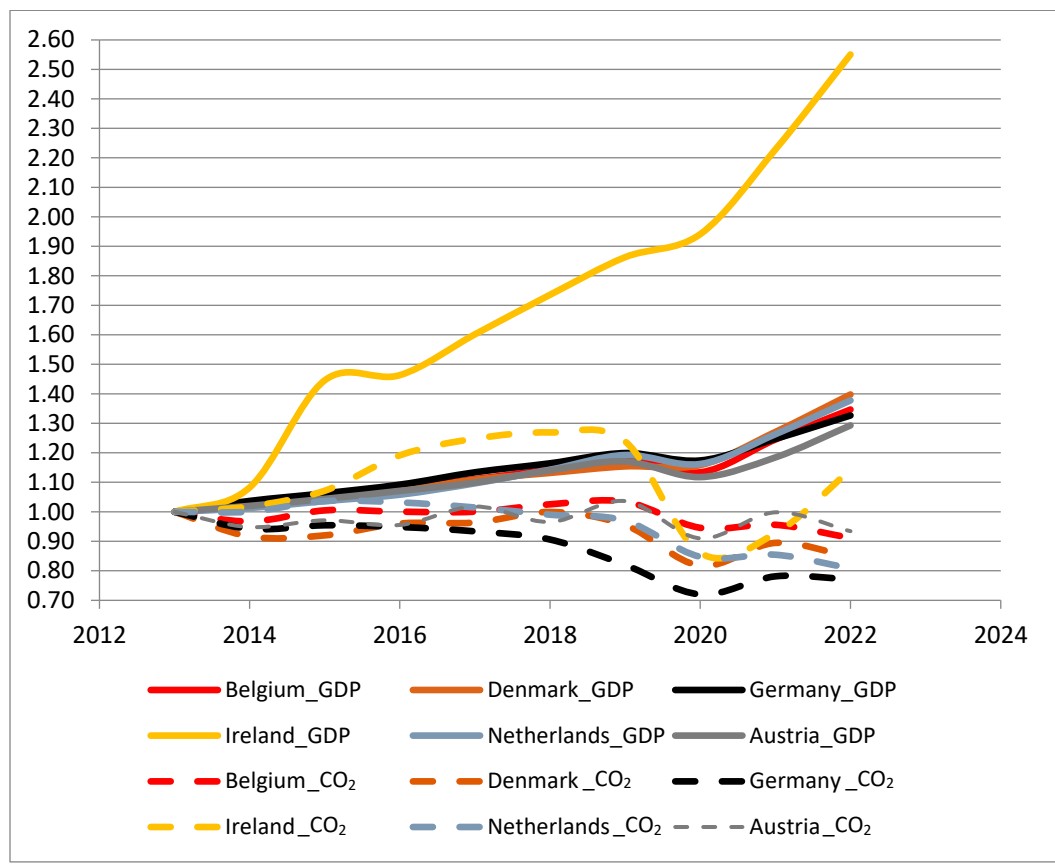

**Figure 6.** Analysis of changes in the level of GDP and $CO_2$ indices in the years 2013–2022 up to the base year 2013—cluster 5, source: own study.

The countries classified in the first cluster are characterized by the highest share of renewable sources in energy balances compared to other member countries. The level of gross domestic product increased steadily in the years examined in relation to 2013, reaching the levels of Luxembourg by 30%, Finland by 28%, and Sweden by 16% in 2022. The highest decrease in emissions in 2022 in relation to 2013 levels is characterized by Finland reaching 32%, then Sweden at 16%. A different situation characterizes Luxembourg, which in 2022 reaches a level slightly higher than in 2013, i.e., it records an increase in emission levels by 0.6%; this fact should be assessed as a negative phenomenon and at the same time, the importance of the selection of the GDP variable should be emphasized as it is justified in analyses in connection with the emission level because even countries with a high standard of living do not achieve significant reductions in the emission levels of their economies.

The countries classified in the second cluster are characterized by a significant share in the nuclear energy mix, i.e., France above 75%, Slovenia above 35%, and Romania close to 20%. These countries use renewable sources to a large extent: France nearly 20%, Slovenia 30%, Latvia 50% (the rest is natural gas), and Romania almost 40%. In the countries surveyed, there was a similar reduction in emission levels of close to 20%. Countries achieved an increase in gross domestic product, which indicates the relative independence of the GDP level from $CO_2$, and this fact should be assessed positively (at the same time, it should be noted that the economic transformation in Latvia and Romania in the field of heavy industries, which had a significant impact on reducing emissions).

Countries classified in the third cluster are characterized by a high level of energy dependence in terms of imported natural gas. Greece achieved the highest emission reduction of 25%. The remaining countries achieved a slight decrease, i.e., Croatia and Portugal nearly 15%, Italy 10%, Hungary only a decrease of 2%, while Lithuania recorded

an increase of 20%—however, this country achieved the highest GDP growth and its value was doubled compared to 2013.

Countries classified in the fourth cluster are characterized by a high share of coal in electricity production. This group includes countries that should change their energy balance in order to reduce their dependence on coal, which generates the highest level of greenhouse gas emissions. In Estonia, as a result of industrial transformation and limiting the share of high-energy-intensive sectors, emissions were reduced by 50%. The Czech Republic achieved a reduction of 11%, and Slovakia achieved 17%. Poland and Bulgaria, in comparison to 2013, recorded a slight increase of 0.3% and 0.5%, respectively, but it should be noted that the transformation of heavy industries took place much earlier. Poland, which is the country with the highest share of hard coal and lignite in the energy balance compared to other European Union countries, will be analyzed in detail in the second part of the article. The decarbonization process in Poland is a very complex process and takes place over a long time because Poland has very rich hard coal deposits and is one of the countries with the highest level of energy independence in the entire EU. However, ecological factors require further transformations in the energy system to achieve the assumed climate goals. The biggest challenge for Poland is the implementation of renewable sources in the required 20% of the energy balance.

The countries classified in the fifth cluster belong to the group of countries with a high share of renewable fuels in their energy mixes. At the same time, these are countries with a very high standard of living and, therefore, have the highest levels of GHG emissions per person. In the years examined, there is a constant high increase in GDP, reaching levels of nearly 40% in 2022 compared to 2013. The highest decrease in emission levels was achieved by Germany, and the reduction amounted to 27%.

We point out that Figures 1–5 indicate a detailed analysis of $CO_2$ emission levels in relation to GDP levels for each country individually. The units of measurement were reduced from the index levels in each subsequent year of the analysis to the base levels in 2013, which significantly enabled the interpretation of the results to prove H1 assumed in the study. Therefore, as shown in the research Figures 1–5, the vast majority of selected European Union countries exhibited the phenomenon of decoupling, i.e., separation of the value of gross domestic product from the value of emissions in the examined years, which confirms their potential to achieve the assumed goals of climate neutrality.

The effects of separating economic growth from the emission level of economies in the surveyed European Union countries were possible to achieve, among others, by increasing energy efficiency and using low-emission energy sources [22].

In the next stage of the research, the relationship between GDP and $CO_2$ was calculated for each Member State. The study results characterize the countries' potential for energy transformation in line with the set goals of the green economy and achieving energy neutrality by 2050. The research results are presented in Table 1.

The article presents an analysis of statistical data regarding the level of economic growth and carbon dioxide emission levels. Statistical data were indexed to the 2013 level to obtain relative levels of increases or decreases in the studied variables. The results are presented in Table 1. The study clearly showed that in the European Union, by implementing emission control measures, the vast majority of countries achieved positive emission reduction effects in 2022 compared to the base year 2013. The largest decreases in $CO_2$ emissions compared to 2013, ranging from 0.5 to 0.7, were achieved by Estonia, Malta, Finland, Greece, and Germany. Decreases ranging from 0.8 to 0.9 were achieved by the following countries: The Netherlands, Romania, Portugal, Slovakia, Croatia, Denmark, Sweden, France, Slovenia, Latvia, Czechia, Italy, Belgium, Spain, Austria, and Hungary.

Emission levels comparable to 2013 were achieved by the following countries: Poland, Bulgaria, Luxembourg, Cyprus, Ireland, and Lithuania. Importantly, countries with a comparable level of emissions in 2012 achieved a constant increase in the level of gross domestic income between 2012 and 2013, which means that there is a gradual separation

of the level of economic growth from the level of emission intensity of economies, which should be assessed positively.

The Pearson correlation coefficient between the GDP and $CO_2$ variables showed levels of negative dependence ranging from $-0.8$ to 0 in the following countries: Romania, The Netherlands, Finland, Germany, Croatia, Estonia, Malta, Slovakia, Czechia, Denmark, Sweden, France, Portugal, Belgium, Latvia, Greece, Slovenia, Italy, Bulgaria, Spain, Hungary, and Austria. This phenomenon should be interpreted positively because it means that decoupling is taking place in these countries, which means that GDP growth is separated from the emission level of the economy. In the case of strict ecological limits on the economy, the higher the level of decoupling, the greater the country's potential to achieve energy transformation to achieve an environmentally neutral economy. Only in the countries Ireland (0.0106), Luxembourg (0.0422), Cyprus (0.2695), Poland (0.4019), and Lithuania (0.6481) was there a positive relationship between GDP and the level of $CO_2$ emission, which means that economic growth affects the increase in emission levels.

Based on the presented analysis, it can be expected that the surveyed countries achieving a negative level of correlation between GDP and $CO_2$ will continue their economic growth while emitting fewer emissions.

On the other hand, to achieve carbon neutrality, all global greenhouse gas emissions will need to be offset by carbon sequestration. A carbon sink is any system that absorbs more carbon dioxide than it emits. The main natural carbon sinks are soil, forests, and oceans. According to estimates, natural sinks remove from 9.5 to 11 gigatons of $CO_2$ per year. Annual global $CO_2$ emissions reached 37.8 Gt in 2021. To date, no artificial carbon sinks have been able to remove $CO_2$ from the atmosphere on the scale needed to combat global warming. Carbon dioxide absorbed by natural sinks, such as trees, is released into the atmosphere as a result of forest fires or deforestation. That is why reducing greenhouse gas emissions is so important to achieve climate neutrality [28,29]. The second way to reduce $CO_2$ emissions is to increase energy efficiency and eliminate fossil fuels from the energy mixes of EU member states by 2050 [23–25].

The second part of the research in the article concerns Poland's special situation compared to other European Union countries. Poland is the only country whose energy balance is mainly based on hard coal and lignite, which is over 80%. The research draws attention to the fact that member states differ significantly in terms of the structure of fuels used to produce electricity. This argument should be the main reason for differentiating the transition periods for those economies that are most dependent on coal, but the financial capabilities of these countries should also be taken into account so that the energy transformation does not have a negative impact on the standard of living of the societies of these countries. In this part of the research, a model based on differential equations was used to simulate the impact of GDP, energy intensity, and the share of biofuels on temperature and $CO_2$ concentration until 2030, using the example of data for Poland. The aim of this analysis is to answer the question of whether the energy transformation in the country will achieve the assumed emission reduction goals by 2030. Poland, like other European Union member states, faces the challenge of achieving climate neutrality by 2050. The Polish economy, based mainly on fossil fuels, especially hard coal and lignite, for many years, faces a particularly difficult task of transformation towards climate neutrality. Coal, as the primary energy source, played a key role in the Polish energy mix, which transmitted high levels of $CO_2$ emissions. The challenge Poland faces is to gradually move away from coal and other fossil fuels towards cleaner energy sources, such as renewable energy while ensuring energy security and the competitiveness of the economy. In the context of analyzing GDP and $CO_2$ data, Poland must pay special attention to the decarbonization of its economy. The Pearson correlation coefficient between GDP and $CO_2$ emissions, which is 0.4019, indicates that economic growth in Poland is still linked to $CO_2$ emissions. To achieve climate goals, Poland must accelerate its energy transformation efforts, which will include increasing the share of renewable energy in the energy mix, improving energy efficiency, and developing low-emission technologies. Decarbonizing the economy requires complex

and long-term actions that will impact various sectors, from energy through industry to transport and construction. Poland, as a country historically dependent on coal, must face particular challenges, such as the restructuring of coal mining, the retraining of workers, and the development of alternative, sustainable sources of income for mining regions. At the same time, Poland has the opportunity to use the energy transformation as an impulse for innovation and the development of new economic sectors. Investments in renewable energy sources, such as wind energy, solar energy, or biomass, can contribute to the creation of new jobs and increase the competitiveness of the economy in the international arena. To sum up, Poland faces the challenge of balancing the need for economic growth with the need to reduce $CO_2$ emissions. The transition to a low-carbon economy will require significant investment and structural change, but it is essential to achieving climate goals and ensuring the country's future sustainable development.

For this purpose, a mathematical model was developed to analyze and predict future environmental conditions in accordance with the adopted energy policy until 2040. This model takes into account data such as greenhouse gas emissions, energy intensity, the share of biofuels in transport fuels, and population and GDP forecasts.

This model is based on differential equations that simulate the impact of GDP, energy intensity, and the share of biofuels on temperature and $CO_2$ concentration until 2030:

Dependency of temperature change:

$$\text{dtemp\_dt} = 0.01 \times \text{emission\_factor} - 0.02 \times (\text{temperature} - 14.0) \tag{1}$$

where

dtemp_dt—represents the rate of temperature change over time,
emission_factor—is a function of GDP, energy intensity, and the share of biofuels,
temperature is the current temperature.

Dependency of $CO_2$ concentration change:

$$\text{dco2\_dt} = \text{emission\_factor} - 0.03 \times \text{co2\_concentration} \tag{2}$$

where

dco2_dt—represents the rate of change of $CO_2$ concentration over time,
co2_concentration—is the current $CO_2$ concentration in the atmosphere.
and:

$$\text{emission\_factor} = \text{gdp} \times \text{energy\_intensity} \times (1 - \text{biofuels\_share}) \tag{3}$$

where

gdp—is the gross domestic product,
energy_intensity—is the energy intensity,
biofuels_share—is the share of biofuels.

This model takes into account that the increase in temperature is driven by emission factors but is also mitigated by natural processes aiming to restore thermal equilibrium (hence the term ($-0.02 \times$ temperature $- 14.0$)). Similarly, the $CO_2$ concentration in the atmosphere increases due to emissions but is reduced by natural $CO_2$ absorption processes (hence the term ($-0.03 \times$ (co2_concentration)).

In the construction of the differential equations utilized within the model, the coefficients represent critical factors that have been meticulously calibrated against empirical data and scientific research [26–33] to accurately reflect the complex dynamics of climate change. For instance, the coefficient of 0.01 applied to the emission factor within the temperature change equation encapsulates the temperature's sensitivity to emissions, as determined by extensive analysis of historical climate data and projections from established climate models. This coefficient quantifies the expected rise in global temperature corresponding to a specified increase in greenhouse gas emissions. Concurrently, the coefficient of $-0.02$ in the same equation signifies the rate at which the temperature gravitates towards a baseline

or equilibrium state, here represented by 14.0 degrees Celsius. This figure is derived from the Earth's natural energy balance, encapsulating the combined effect of anthropogenic and natural influences on temperature fluctuations.

Similarly, within the $CO_2$ concentration change equation, the coefficient of $-0.03$ is indicative of the natural sequestration rate of atmospheric $CO_2$, encompassing processes such as oceanic uptake and terrestrial biosphere absorption. The determination of this coefficient is grounded in scientific studies that quantify the rate at which these natural sinks mitigate atmospheric $CO_2$ levels.

It is imperative to acknowledge that these coefficients are simplifications, serving as proxies for the multifaceted interactions within the Earth's climate system. The model's predictive capacity hinges on the precise calibration of these coefficients to align with observed data, thereby ensuring the reliability of its future climate projections. The selection of these specific values is predicated on their ability to provide the closest fit to historical observations of temperature and $CO_2$ concentrations.

The results obtained from the models are shown in Figures 7 and 8.

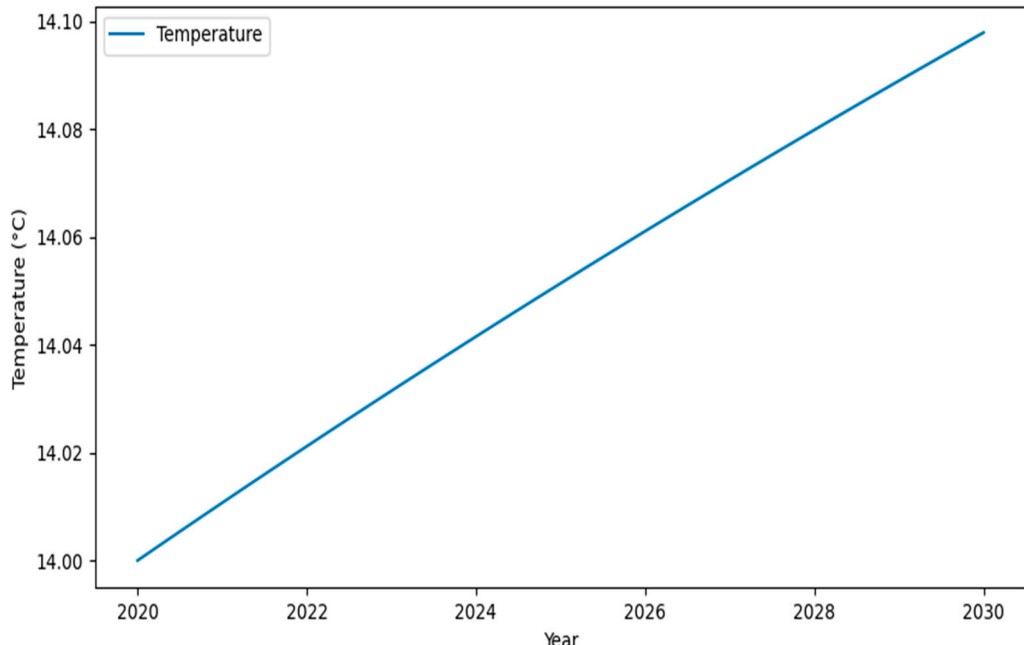

**Figure 7.** Temperature simulation, source: own study.

The temperature graph shows an upward trend; it suggests that the model predicts an increase in average temperatures over the decade. This could be due to high energy intensity and low biofuel share, which implies that energy use is increasing without a corresponding increase in the use of renewable energy sources. A flat or downward trend would suggest that the measures taken to reduce energy intensity or increase biofuels share are effective in mitigating temperature rise.

A downward trend would suggest that the model predicts a successful reduction in $CO_2$ emissions, possibly due to effective policies promoting energy efficiency and biofuels.

The steepness of the curve shows how quickly $CO_2$ levels are changing, and any inflection points could indicate a change in the rate of emissions over time.

The developed climate model for Poland indicates that by 2030, we may notice an increase in average temperature but, at the same time, a decrease in $CO_2$ emissions in the context of GDP growth. This result can be interpreted as a sign of positive changes in the structure of the economy and energy efficiency.

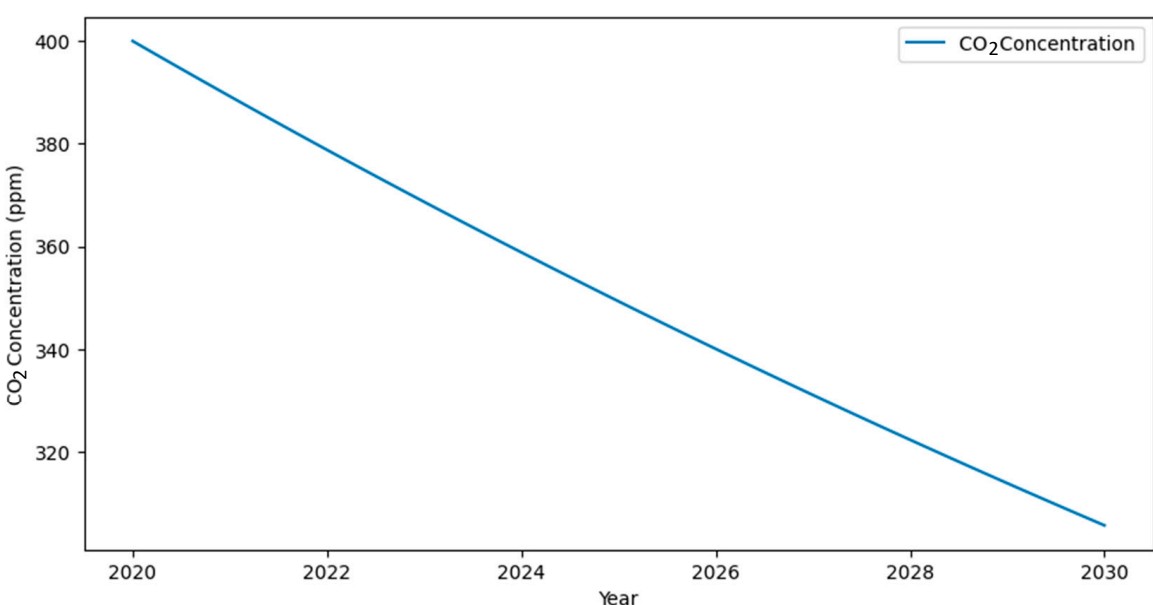

**Figure 8.** $CO_2$ concentration simulation, source: own study.

The increase in temperature, despite the decrease in $CO_2$ emissions, may be due to a climate lag, where previous $CO_2$ emissions continue to affect the climate due to the long lifetime of $CO_2$ in the atmosphere. In other words, current actions may reduce emissions, but the effects of previous actions still persist.

The decrease in $CO_2$ emissions while GDP increases may be the result of several factors:

- Increasing energy efficiency: As economies develop, technologies become more efficient, meaning less energy is needed to produce the same amount of goods and services. This may lead to a reduction in $CO_2$ emissions, even with an increase in GDP;
- Shift to renewable energy sources: Poland can invest in clean technologies such as wind, solar, and bioenergy that replace fossil fuels and help reduce greenhouse gas emissions;
- Decarbonization of the economy: GDP growth can be driven by low-$CO_2$ sectors, such as services and information technologies, which replace traditional, more carbon-intensive industrial sectors;
- Environmental policy and regulation: Poland's introduction of policy measures such as carbon taxes, emissions trading systems, and subsidies for green technologies may result in reduced $CO_2$ emissions, even with economic growth.

## 4. Discussion

Similar research on, among others, the differences between European Union countries in terms of actions taken in the field of energy and climate policy resulting from European Union directives was presented in [34]. In the article, on the basis of the energy policy objectives presented in the EU directives, indicators describing the activities of member states in this area were selected. Then, they were subjected to substantive and statistical verification, leaving six variables. The use of Ward's method allowed for the creation of groups of countries similar in terms of energy and climate policy. Studies have shown that the most favorable situation in terms of the effects of the climate and energy policy can be observed in the current leaders, Sweden and Finland, as well as Estonia. The countries of Central Europe, mainly Poland, are characterized by a high-emission economy accompanied by low expenditure on modern environmentally friendly technologies.

The methodologies adopted by Cluster 1 countries, which are at the forefront of renewable energy integration, align with the findings of [35], who highlight the effectiveness of policy-driven renewable energy adoption in reducing emissions. Our results corroborate

their conclusion that government incentives play a pivotal role in accelerating the transition to clean energy. Furthermore, the work of [36] on smart grid technologies provides a complementary perspective on the importance of infrastructure modernization, which is a key component of our Cluster 1 analysis. In Cluster 2, our observation of industrial efficiency driving emission reductions is supported by research [37], which documents similar trends in high-efficiency, low-emission economies. Their study emphasizes the role of technological innovation in achieving decoupling, which is consistent with our findings. The transitioning economies of Cluster 3, with their diverse energy mixes, are reflected in the studies by [38], who examine the challenges and opportunities of energy source diversification. Their insights into the role of natural gas as a bridge fuel offer a valuable comparison to our analysis, underscoring the complexity of transitioning from coal dependency. For Cluster 4, our focus on the intersection of energy and agricultural practices finds resonance in the research by [39], who investigate the potential of biogas in agricultural settings. Their positive assessment of biogas for emission reduction complements our methodology, highlighting the sector's untapped potential. Lastly, the gradual transition plans of Cluster 5's fossil fuel-dependent economies are paralleled in the longitudinal study by [40], which tracks the progress of similar economies over a decade. Their findings on the socioeconomic implications of transition strategies provide a crucial context for our analysis, emphasizing the need for a just and equitable approach.

As demonstrated in the EU countries, in most of the surveyed countries, there was a phenomenon of decoupling economic growth from the emission level of the economies. The examined period covered the last 9 years. Conclusions based on such a time range are subject to a high degree of uncertainty as to the long-term trend until 2050. However, research indicates the potential of countries that have already made pro-environmental changes in the economies of the European Union to achieve stringent environmental goals. An important aspect that may also be important in the presented research area is the volume of imports of goods and services to EU countries. For many years, we have been observing an increase in the level of imports, which may mean a decline in production in regions subject to stringent ecological requirements in favor of an increase in production in regions where limits do not apply. In such a case, the possibility of only apparent decoupling should be considered, which means that environmental pollution is generated in other regions of the world, including Asia, and the finished products are consumed in EU countries. The article does not exhaust the broad subject of the energy transformation of the European Union countries, which sets ambitious ecological goals to be achieved in 2050. However, the indicated research areas confirm the changes taking place in the economies of the member states and clearly confirm the ongoing pro-environmental transformations and trends. Most countries are on a path to achieving emission limits while maintaining the potential to maintain economic growth. As indicated in the studies, countries differ significantly in terms of the structure of electricity production, which is the main determinant of economic growth. The transformations and the pace of ongoing changes must take into account the economic capabilities of individual countries so that the effects of the transformation on economies dependent on fossil fuels have the hallmarks of a just transformation. The presented research material and areas of analysis are also signals for further research in the field of forecasted trends and scenarios of changes.

## 5. Conclusions

Achieving very stringent climate goals has been introduced in the European Union countries as a leading trend on a global scale. Today, a clear definition of the possibility of achieving this goal by all member states seems to be burdened with a very high degree of uncertainty. The authors of the study set the goal of this study as an answer to the question of whether the member states have entered the path of energy transformation, thereby enabling the member states to achieve the assumed climate goal in 2050. Based on the presented analyses, it was determined that most countries, despite the diversified level of energy balance structure, have significant potential to achieve energy neutrality

in the future. To summarise, the model suggests that Poland may experience economic growth that goes hand in hand with the energy transition and changes in the structure of the economy, leading to temperature increases that are, however, lower than those resulting from $CO_2$ emissions, thanks to effective climate protection actions. This shows that economic development does not have to conflict with environmental goals, provided appropriate policies and investments in sustainable technologies are in place.

**Author Contributions:** Conceptualization and methodology A.B. and A.M.; software, A.M. and N.S.M.; formal analysis, A.B. and A.M.; writing—original draft preparation, A.M.; writing—review and editing, A.B. All authors have read and agreed to the published version of the manuscript.

**Funding:** This research was funded by Statutory Research BK2024_RG1_RG3.

**Institutional Review Board Statement:** Not applicable.

**Informed Consent Statement:** Not applicable.

**Data Availability Statement:** The data presented in this study are available on request from the corresponding author.

**Conflicts of Interest:** The authors declare no conflicts of interest.

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
