# Peer review of "Assessment of the Potential of European Union Member States to Achieve Climate Neutrality"

_sustainability, doi:10.3390/su16031311_

Round 1

Reviewer 1 Report

Comments and Suggestions for Authors

Dear authors, thank you for your really actual article to the very actual theme as the climate neutrality is. I have a few comments:

- please there is not readeble in your figures too small letters
- also in the figures there is no described x-axis and y-axis
- what is missing is in the part Discussion where should be mentioned also the results of another authors which should be compared with your results. This could also support the part of Literature, while I think there is too little used literature sources
- part Study methods please provide more deeply described methods used in your analysis

Also I think that the method of Dependency of temperature change should be described in part of used methids in you artcile

Author Response

Thank you for your valuable comments, all of them have been taken into account

Reviewer 2 Report

Comments and Suggestions for Authors

The topic addressed in this review article is very current and of utmost importance in the scientific community, encompassing various sectors of society and scientific research.

Despite the article having a very interesting topic, I have several questions and reservations about the value of this article in different aspects, such as:

  • - Where are the models used in this study based? In an initial section, is there no analysis of CO2 emissions for each country and a comparison with its economic development? Wouldn't it be expected that with policies to reduce CO2 emissions, countries would continue their normal development while emitting less CO2?

  •  
  • - Couldn't this reduction in CO2 be achieved in different ways? Has it not been identified how member states individually contributed to the reduction of CO2? Wouldn't it be interesting to do so?

  •  
  • - Could the authors analyze the investment that each member state made in reducing emissions, normalized by its GDP (or another parameter)? It seems to me that the analysis done was quite simple and lacked significant conclusions for the readers.

In this way, I suggest that the authors add a chapter on the methodologies used by the member states (for example, separated by clusters) to reduce CO2 emissions (It can also be separated by end-use). It would also be interesting to discuss EU projects developed to achieve the goals and, in the end, indicate the paths that still need to be taken.

Author Response

(The authors gave the same response as above.)

Reviewer 3 Report

Comments and Suggestions for Authors

The reviewed manuscript is at a good level in terms of the originality of the solutions presented and the final conclusions formulated, and will probably meet with the interest of readers, and its results can be used for further research on the projected trends and scenarios of changes to achieve the assumed climate goal in the EU in 2050.

The manuscript is mainly descriptive, and fragmentary numerical research concerns only the case of Poland, for which a mathematical model has been developed to analyze and predict future environmental conditions, in accordance with the adopted energy policy of this country until 2040. The descriptive part is comprehensive and clear, but the mathematical model requires a broader description.

However, the presented manuscript requires some changes, primarily in order to improve its readability, i.e. additions and corrections should be made to the presented text, especially in Chapter 1. Introduction and in Chapter 4 Results of analysis.

Below are selected detailed remarks:

1) it is recommends a broader description of the implementation of the Carbon Leakage Instrument, which sets a greenhouse gas emission fee for imported goods, as a very important element of the EU's energy policy (in Chapter 1),

2) It is recommended to describe the mathematical model presented in Chapter 4 in more detail,

3) the presented text requires editorial corrections, for example:

- line 275 should be a continuation of line 274

- line 280 year 20122 should be corrected.

Comments on the Quality of English Language

Minor editing of English language required

Author Response

(The authors gave the same response as above.)

Round 2

Reviewer 2 Report

Comments and Suggestions for Authors

-